# Geobarrettin D, a Rare Herbipoline-Containing 6-Bromoindole Alkaloid from *Geodia barretti*

**DOI:** 10.3390/molecules28072937

**Published:** 2023-03-24

**Authors:** Xiaxia Di, Ingibjorg Hardardottir, Jona Freysdottir, Dongdong Wang, Kirk R. Gustafson, Sesselja Omarsdottir, Tadeusz F. Molinski

**Affiliations:** 1Faculty of Pharmaceutical Sciences, University of Iceland, Hagi, Hofsvallagata 53, IS-107 Reykjavik, Iceland; 2Department of Immunology, Landspitali—The National University Hospital of Iceland, IS-101 Reykjavik, Iceland; 3Faculty of Medicine, Biomedical Center, University of Iceland, Vatnsmyrarvegur 16, IS-101 Reykjavik, Iceland; 4Molecular Targets Program, Center for Cancer Research, National Cancer Institute, Frederick, MD 21702, USA; 5Department of Chemistry and Biochemistry, Skaggs School of Pharmacy and Pharmaceutical Sciences, University of California, San Diego, CA 92093, USA

**Keywords:** 6-bromoindole, herbipoline, *Geodia barretti*, anti-inflammatory activity, dendritic cells

## Abstract

Geobarrettin D (**1**), a new bromoindole alkaloid, was isolated from the marine sponge *Geodia barretti* collected from Icelandic waters. Its structure was elucidated by 1D, and 2D NMR (including ^1^H-^15^N HSQC, ^1^H-^15^N HMBC spectra), as well as HRESIMS data. Geobarrettin D (**1**) is a new 6-bromoindole featuring an unusual purinium herbipoline moiety. Geobarrettin D (**1**) decreased secretion of the pro-inflammatory cytokine IL-12p40 by human monocyte derived dendritic cells, without affecting secretion of the anti-inflammatory cytokine IL-10. Thus, compound **1** shows anti-inflammatory activity.

## 1. Introduction

The indole nucleus is an important element of many natural and synthetic molecules possessing significant biological activities. Indole has been termed a “privileged structure” in drug discovery and a common starting point for drug development or lead optimization [1,2]. Marine indole alkaloids comprise a large and complex class of natural products; most marine-derived indole metabolites are halogenated by bromine [2,3,4]. The presence of halogen substituents on the indole ring profoundly influences biological activity [2,3]. Interestingly, the Br substituent generally resides at C-5, less commonly at C-6, or at both C-5 and C-6 [4]. Additional modifications of the bromoindole core include C-substitutions by O, C (often prenyl) and *N* groups, pyrimidyl, 2-aminopyrimidyl groups or more complex polycyclic ring systems [5,6,7,8,9,10,11]. Bromoindoles have been reported to have anti-inflammatory, antibacterial, antifungal, antitumor, antioxidant, antifouling, and antiplasmodial activities [1,3,4,12,13].

The marine sponge *Geodia barretti* is the source of bromoindole alkaloids [14,15,16,17,18] and several *N*-methylated nucleosides [19]. In our previous study, three new 6-bromoindole derivatives were isolated from *G. barretti* collected in Icelandic waters [20]; of these, geobarrettins B and C exhibited anti-inflammatory activity [20]. As part of our ongoing investigation to find novel anti-inflammatory compounds, we report, here, the bromoindole geobarrettin D (**1**) (Figure 1, as the TFA salt) with potential anti-inflammatory effect measured by decreased pro-inflammatory cytokine secretion of human monocyte-derived dendritic cells (DCs).

## 2. Results

The lyophilized sponge *G. barretti* was extracted with CH_2_Cl_2_/MeOH (*v*/*v* 1:1). After removal of solvent, the resulting crude extract was resuspended in MeOH/H_2_O (9:1) and solvent-partitioned into five fractions of increasing polarity (hexane, CHCl_3_, CH_2_Cl_2_, *n*-BuOH, and H_2_O) using a modified Kupchan method [21,22]. The CHCl_3_ and CH_2_Cl_2_ fractions were combined and purified by RP C18 HPLC to afford the 6-bromoindole derivative, geobarrettin D (**1**, Figure 1).

### 2.1. Structural Elucidation

The HRMS of geobarrettin D (**1**) exhibited molecular ion isotopomers *m*/*z* 458.1343/460.1316 ([M]+) in a 1:1 ratio, indicating the presence of one Br and a molecular formula C_20_H_25_^79^BrN_7_O, corresponding to 12 degrees of unsaturation. The ^13^C NMR data (Table 1, see Appendix A) showed 18 signals which were matched to the C content of the molecular formula, including five methyls (*δ*_C_ 54.8 (×3), 36.2, and 32.1), one sp^3^ methylene (*δ*_C_ 69.3), one aliphatic methine (*δ*_C_ 45.4), five aromatic methines (*δ*_C_ 140.1, 125.9, 124.2, 120.8, and 115.9), four quaternary aromatic carbons (*δ*_C_ 139.1, 125.2, 117.0, and 113.4), four quaternary heteroatom-bonded sp^2^ carbons (*δ*_C_ 155.0, 154.9, 151.0, and 110.0) (Table 1). The most intense MS peak at *m*/*z* 399.0603/401.0585 ([M-*N*(CH_3_)_3_]^+^) (Appendix A) was derived from a neutral loss of trimethylamine (–*N*(CH_3_)_3_). IR bands (3254, 1182, and 1131 cm^−1^) implied the presence of OH and/or NH functionalities. The ^1^H NMR data of **1** in CD_3_OD (Table 1) exhibited signals due to five aromatic protons, five *N*-methyl groups, [*δ*_H_ 4.11(3H, s, 11-Me), 3.91(3H, s, 12-Me), and 3.27 (9H, s, 1″-*N*Me)], a methine proton, and a methylene group. Three aromatic signals at *δ*_H_ 7.65 (1H, d, *J* = 8.5 Hz, H-4′), 7.60 (1H, d, *J* = 1.6 Hz, H-7′) and 7.24 (1H, dd, *J* = 8.5, 1.6 Hz, H-5′) indicated the presence of a 1,2,4-trisubstituted benzene ring. The ^1^H NMR spectrum, recorded in D_2_O/H_2_O (1:9), showed the presence of a downfield exchangeable proton (*δ*_H_ 10.60), which is diagnostic of an *N*H proton in an indole ring, and confirmed by a weak coupling (*J* = 2.0 Hz) to the isolated aromatic proton at *δ*_H_ 7.53 in the pyrrole ring. Correlations observed in the HMBC spectrum (Figure 2) allowed the definement of the substitution pattern and NMR assignments of the indole: H-4′ to C-3′, C-6′, and C-7a′, from H-5′ to C-3a′ and C-7′, and from H-7′ to C-5′ and C-3a′. H-2′ also showed a correlation to C-3′, C-3a′, and C-7a′. The *N*H exchangeable proton H-1′ showed correlations to C-2′, C-3′, C-3a′, and C-7a′. The indole assignment was supported by the presence of several bands in the UV-vis spectrum (*λ*_max_ 228, 261, and 287 nm).

Br-substitution at C-6′ was deduced by a comparison of the chemical shifts of the aromatic carbons of related 6-bromoindole alkaloids [16,18,23]. Thus, geobarrettin D (**1**) was defined as a 3-substituted 6-bromoindole alkaloid. The ^1^H-^1^H COSY correlations from H-3″ (*δ*_H_ 6.05 (1H, t, *J* = 6.3 Hz)) to H-2″ (*δ*_H_ 4.07 (1H, dd, *J* = 13.7, 5.9 Hz); 4.15 (1H, dd, *J* = 13.7, 6.8 Hz)) and HMBC correlations of (CH_3_)_3_-*N* (*δ*_H_ 3.30 (9H, s))/C-2″ (*δ*_C_ 69.3), H-2″/C-3″ (*δ*_C_ 45.4) and H-3″/C-2″ indicated the presence of 2,2-disubstituted *N*,*N*,*N*-trimethylethanaminium group; further support of the connectivity of 6-bromo-indol-3-yl moiety and the *N*,*N*,*N*-trimethylethanaminium group were provided by additional HMBC correlations: H-3″/C-3′ (*δ*_C_ 113.4), H-3″/C-2′ (*δ*_C_ 125.9), H-3″/C-3a′ (*δ*_C_ 125.2), and H-2″/C-3′ (Figure 2).

The balance of the molecular formula C_20_H_25_^79^BrN_7_O of geobarrettin D (**1**), C_7_N_5_H_8_O, after accounting for the 6-bromoindole and 2,2-disubstituted *N*,*N*,*N*-trimethylethanaminium moieties, required another six degrees of unsaturation. The HMBC cross-peak H-3″/C-2 (*δ*_C_ 154.9) revealed that the C_7_N_5_H_8_O unit was connected to C-3″ through a C-*N* bond, which explains the downfield chemical shift of C-3″ (*δ*_C_ 45.4). Analysis of the ^13^C NMR data revealed the seven remaining carbons as non-protonated sp^2^ carbons with chemical shifts of *δ*_C_ 155.0, 154.9, 151.0, 140.1, 110.0 and two sp^3^ carbons *δ*_C_ 36.2, 32.1 (Table 1). The ^1^H NMR chemical shift of the non-exchangeable *δ*_H_ 9.01 lacked an expected cross-peak in the HSQC spectrum, but strong symmetric ‘satellite peaks’ appearing in the HMBC, centered on δ_C_ 140.1, were due to ^1^*J*_CH_ ‘breakthrough’ (^1^*J*_H8-C8_ = 220 Hz) [24,25]: the large magnitude is consistent with a five-membered heterocycle [26]. Long-range correlations were also seen from H-8 [*δ*_H_ 8.81 (1H, s, H-8)] to two *N*-methyl groups (δ_C_ 36.2 and 32.1 ppm) in addition to C-4 and C-5 (δ_C_ 151.0 and 110.0, respectively). These data are reconciled by an *N*,*N*-dimethyl imidazole ring.

H-detected ^15^N-heteronuclear 2D NMR experiments (^1^H−^15^N HSQC and ^1^H-^15^N HMBC in D_2_O/H_2_O, 1:9) were also recorded. The correlations from H-8 [*δ*_H_ 8.81 (1H, s, H-8)] to *N*-9 (*δ*_N_ 157.3), *N*-7 (*δ*_N_ 156.4), from CH_3_-12 (*δ*_H_ 3.91 (3H, s)) to *N*-9, from CH_3_-11 (*δ*_H_ 4.11 (3H, s)) to *N*-7 in ^1^H−^15^N HMBC and ^1^H−^13^C HMBC spectra further supported a *N*,*N*-dimethyl imidazolinium ring. The latter partial structures, together with the last two degrees of unsaturation, were assembled with the remaining quaternary C and three N atoms to complete an *N*-quaternized guanininium nucleobase. This hypothesis was supported by the H-8/C-6 ^4^*J*_CH_ correlation and the downfield chemical shift of C-4, and comparisons of ^13^C shifts of **1** with published data for similar purine bases, e.g., herbipoline (7,9-dimethyl-2-(*N*-amino)guaninium) [25,27,28,29,30] with the same C-2″–*N*-10 bond. The NMR data are in good agreement with other natural alkylpuriniums: 7,9-dimethyl-2-(*N*-methyl)guaninium chloride [25] and *N*,*N*-dimethyl-1,3-dimethylherbipoline [30].

Although **1** is chiral, the weak optical activity ([α]_23_^D^ +2 (*c* 0.4, MeOH)) suggests a near-racemic mixture.

### 2.2. Anti-Inflammatory Activity

When DCs were matured and activated in the presence of geobarrettin D (**1**), secretion of the pro-inflammatory cytokine IL-12p40 was diminished by 48%, whereas secretion of the anti-inflammatory cytokine IL-10 was not affected (Figure 3). On balance, these results indicate an overall anti-inflammatory effect of geobarrettin D (**1**).

## 3. Discussion

Marine sponges have proven to be a rich source of naturally occurring modified nucleosides: these exist as free bases, nucleotides, and within polynucleotides [31,32]. The first natural purinium salt found in nature, herbipoline, was isolated from the sponge *Geodia gigas* [32]. Subsequently, several related herbipoline salts were characterized from tropical marine sponges: l-methylherbipoline from *Jaspis* sp. [25]], 1-methylherbipoline salts of halisulfate-1 and suvanine from *Coscinoderma mathewsi* [27] and suvanine (*N*,*N*-dimethyl-1,3-dimethylherbipoline salt) from *Coscinoderma* sp. [30]. A literature survey revealed that **1** is the first herbipoline-containing indole [32], and a rare bis-quaternized alkaloid. 

Most natural product alkaloids are derived from the aromatic amino acids including tryptophan, tyrosine and phenylalanine. Compound **1** is likely biosynthetically derived from 6-bromotryptamine—an alkaloid known from other marine invertebrates [33]—through a fusion of an oxidized tryptamine intermediate and a guanine equivalent (Figure 4) through a 1,4-conjugate addition, followed by extensive methylation reactions involving *S*-adenosylmethionine (SAM). Oxidation and conjugate addition is necessary and sufficient to explain formation of many C–C and C–heteroatom bonds in alkaloids [34], including **1**. For example, one of us (T.F.M) recently reported two cyclic guanidines, aiolochroiamides A and B, whose formation may also be rationalized by an oxidation–conjugate addition mechanism [35]. It is likely that the oxidation reaction is enzyme-mediated as spontaneous autoxidation seems unlikely, however the subsequent conjugate addition may be spontaneous given the low specific rotation (and therefore enantiomeric excess) observed for **1**. As with other complex highly-methylated quaternized alkaloids, the ordering of *N*-methylation and condensation reactions is uncertain. Further biosynthetic studies are necessary for understanding the biosynthesis of **1**, but these are beyond the scope of this study.

Purines have found antiviral, antibiotic, and anticancer activities [27,31,36,37], and have the potential to regulate myocardial oxygen supply and cardiac blood flow [38]. In addition, evidence supports their role in biological evolution, differentiation, and ecological processes [31]. Purines are also involved in various inflammatory responses which underscores the significant attention given to purine natural products and their synthetic mimetics for the development of anti-inflammatory agents [20,39,40]. 

The anti-inflammatory properties of compound **1** were investigated in an in vitro model of human monocyte-derived DCs [20]. DCs matured and activated in the presence of compound **1** secreted less IL-12p40 than DCs cultured without **1**, whereas **1** had no effect on their secretion of IL-10. The pro-inflammatory cytokine IL-12p40 (one of the two chains that form the structures of IL-12 and IL-23 cytokines) is a major determinant of the differentiation of naïve T cells into Th1 or Th17 phenotypes [41], whereas the anti-inflammatory cytokine IL-10 drives polarization of naïve T cells into a T regulatory phenotype [42]. Thus, suppression of IL-12p40 secretion by DCs in the presence of **1** indicates that geobarrettin D (**1**) has anti-inflammatory activity.

## 4. Materials and Methods

### 4.1. General Procedures

The UV spectrum was recorded on a NanoVueTM spectrophotometer (GE Healthcare Life Sciences, Little Chalfont, UK) with a 0.2 mm path length. Optical rotation was measured on a P-2000 polarimeter (Jasco, Oklahoma City, OK, USA), with a quartz cell (10 mm path length). The infrared spectrum was measured on a Spectrum Two TM FTIR spectrometer (Perkin Elmer^®^, Waltham, MA, USA) of samples as thin films. NMR spectra were recorded on a Bruker Avance 600 spectrometer (Billerica, MA, USA) (^1^H and ^13^C frequencies: 600.13 MHz and 150.76 MHz, respectively) in CD_3_OD and D_2_O/H_2_O (1:9). The residual solvent signals were used as internal references: *δ*_H_ 3.30/*δ*_C_ 49.0 ppm (CD_3_OD) and *δ*_H_ 4.79 (D_2_O). For ^1^H-^15^N 2D NMR spectroscopy, the nominal ^15^N standard was liquid ammonia, NH_3_ (l) (*δ* = 0 ppm). Samples (1.6–6.0 mg) were introduced into Shigemi tubes and their 2D NMR spectra measured as illustrated by the following ^1^H{^13^C} heteronuclear HSQC and HMBC spectra using modifications of the Bruker pulse sequences hsqcedetgpsisp2.4 and hmbcgplpndqf, respectively: Spectra were acquired at near ambient temperature (T = 300.0 K) in the specified solvent with an rf pulse calibrated to 1H π/2 = 8.75 µs, with appropriate gradient field strengths, and dwell times corresponding to ^1^H (F2) and ^13^C (F1) spectral widths of 8417.5 Hz and 33112.6 Hz and centered at *δ*_H_ 7.01 and *δ*_C_ 110.4 ppm, respectively. Accumulated scans (*n* = 8 for HMBC and *n* = 32 for HSQC) for each T1 increment were averaged between a relaxation delay, D1 =1.5 s. The acquired matrix (^1^H and ^13^C, 1024 × 256 increments for HMBC, and 672 × 256 for HSQC) was zero-filled in each dimension to a final size of 2048 × 1024, and processed, after standard apodizations, by Fourier transform.The high-resolution mass spectrum was measured on an Acquity UPLC I-Class System coupled to Xevo G2-XS QTof Mass Spectrometer (Waters^®^, Milford, MA, USA) using Acquity UPLC^®^ HSS T3 column (High Strength Silica C18, 1.8 µm, 2.1 × 100 mm, Waters^®^, (Milford, MA, USA) operating at 60 °C). The MS and MS^n^ spectra were recorded in positive mode and data were acquired using MassLynx^®^ Software (version 4.1, Waters Crop., Milford, MA, USA). VLC chromatography on C_18_ adsorbent (LiChroprep RP-18, 40–63 μm, Merck Inc., Darmstadt, Germany) and Dionex 3000 HPLC system armed with a G1310A isopump, a G1322A degasser, a G1314A VWD detector (210 nm), a 250 × 21.2 mm Phenomenex Luna C18(2) column (5 μm), and a 250 × 4.6 mm Phenomenex Gemini-NX C18 column (5 μm) were conducted for separation and purification of pure compounds. 

### 4.2. Animal Materials

In short, the sponge material *Geodia barretti* was collected in Iceland, identified by. Hans Tore Rapp, University of Bergen (Norway), and deposited at University of Iceland. For a complete description of the samples, see Xiaxia Di et al. [20].

### 4.3. Extraction and Isolation

Frozen sponge was cut into approximately 1 cm^3^ pieces and freeze-dried. The dried tissue was extracted with a mixture of CH_2_Cl_2_:CH_3_OH (*v*/*v*, 1:1) for 3 times (2 L, each for 24 h) at room temperature. The combined CH_2_Cl_2_-MeOH extracts were dried under reduced pressure then the residue (1.8 g) was suspended in MeOH:H_2_O (*v*/*v*, 9:1) and subjected to a modified Kupchan partition, as previously described [21,22], to yield five fractions: hexane (fraction A), chloroform (fraction B), dichloromethane (fraction C), *n*-butanol (fraction D), and H_2_O (fraction E). Using a VLC RP-18 CC (MeOH-H_2_O, 10:90→100:0) technique, the mixture of the fractions B and C was separated into nice fractions (F2.1−F2.9). Fraction F2.2 (75.0 mg) was purified by preparative HPLC (28:72:0.1 CH_3_CN-H_2_O-TFA, 8.0 mL/min) and then re-chromatographed by semi-preparative HPLC (CH_3_CN-H_2_O-TFA, 31:69:0.1) to give geobarrettin D (**1**) (3.3 mg). 

Geobarrettin D (**1**): Light yellowish oil; [α]_23_^D^ +2 (*c* 0.4, MeOH); UV (MeOH) λ_max_ (log ε) nm: 212 (3.90), 228 (4.04), 261 (3.81), 287 (3.65); IR ν_max_ cm^−1^: 3255, 1679, 1607, 1478, 1444, 1385, 1203, 1131; For ^1^H and ^13^C NMR data, see Table 1; HRESIMS *m*/*z* 458.1343 [M]^+^ (C_20_H_25_ON_7_Br, 458.1298).

### 4.4. Maturation and Activation of DCs

DCs were differentiated from human monocytes as previously described [20]. Peripheral blood mononuclear cells (PBMCs) were isolated from peripheral blood obtained from healthy human donors (approval #06-068-V1 by National Bioethics Committee of Iceland (Visindasidanefnd) from 15^th^ of December 2015) by density centrifugation using Histopaque-1077 (Sigma-Aldrich, Munich, Germany). Then CD14^+^ monocytes were isolated from the PBMCs using magnetic cell sorting and CD14 Microbeads (Miltenyi Biotech, Bergisch Gladbach, Germany). The CD14^+^ monocytes were cultured at 5 × 10^5^ cells/mL in RPMI 1640 medium, supplemented with 10% fetal calf serum and 5% penicillin/streptomycin (all from Gibco^®^, Thermo Fisher Scientific, UK) in 48 well tissue culture plates for seven days. In order to differentiate CD14^+^ monocytes into immature DCs, IL-4 at 12.5 ng/mL and GM-CSF at 25 ng/mL (both from R&D Systems, Bio-Techne, Abingdon, England) were added to the cells. After seven days the monocytes had differentiated into immature DCs which were harvested and cultured for 24 h in 48 well tissue culture plates at 2.5 × 10^5^ cells/mL. The immature DCs were matured and activated by culturing them with IL-1β at 10 ng/mL, TNF-α at 50 ng/mL (both from R&D Systems), and lipopolysaccharide (LPS) at 500 ng/mL (Sigma-Aldrich, Munich, Germany). Geobarrettin D (**1**) was dissolved in DMSO and added to the DCs at 10 µg/mL at the same time as the cytokines and LPS. DMSO was used as a control. After 24 h the mature and activated DCs were harvested and the concentrations of IL-12p40 and IL-10 in the supernatants were measured by sandwich ELISA using DuoSets from R&D Systems according to the manufacturer’s protocol. The results are expressed as a secretion index (SI). Data are presented as the mean values ± SEM, *n* = 3. As the data were not normally distributed, Mann–Whitney U test was used to determine statistical differences between the groups (SigmaStat 3.1, Systat Software, San Jose, CA, USA) and *p* < 0.05 was considered as statistically significant.

## 5. Conclusions

Geobarrettin D (**1**) is a newly detected bromoindole alkaloid possessing an unusual brominated and fused-herbipoline-dimethylguaninium heterocycle. Geobarrettin D (**1**) decreased DC secretion of the pro-inflammatory cytokine IL-12p40, indicating its potential as an anti-inflammatory agent. 

## Figures and Tables

**Figure 1 molecules-28-02937-f001:**
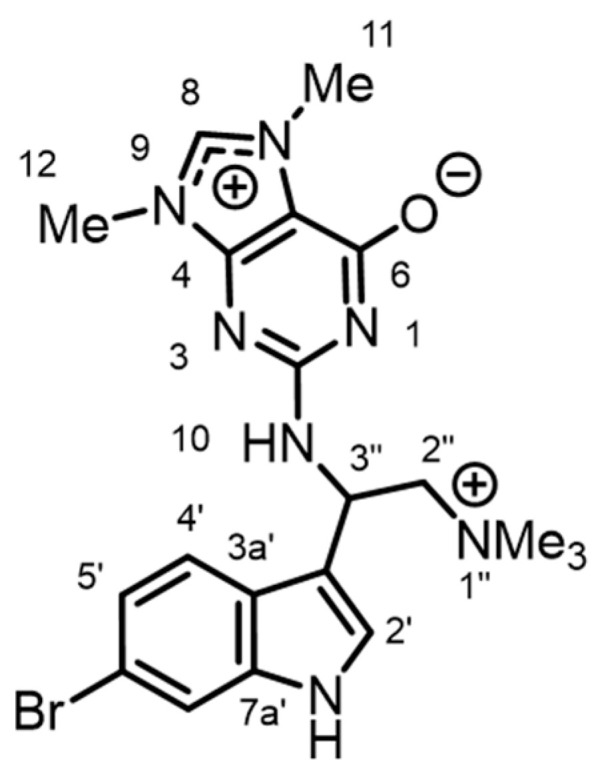
Structure of geobarrettin D (**1**).

**Figure 2 molecules-28-02937-f002:**
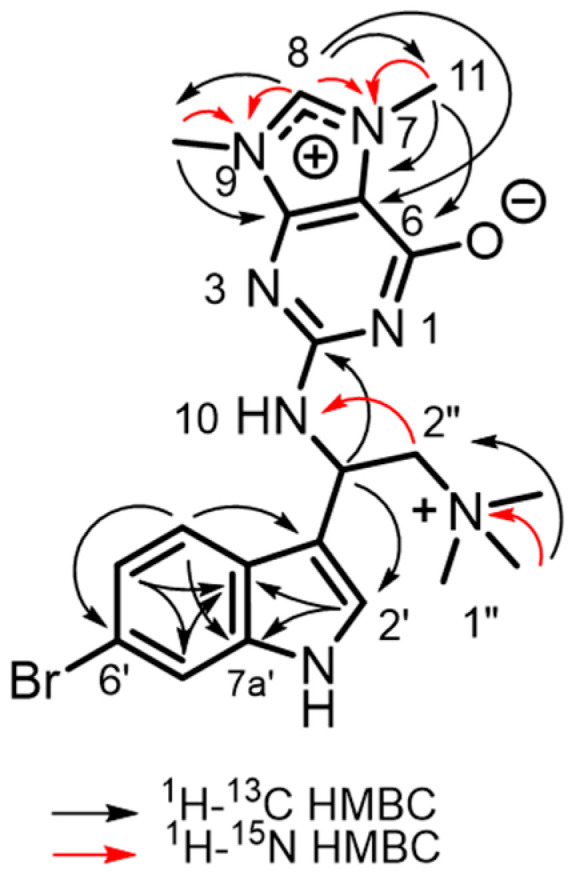
Key ^1^H-^13^C HMBC and ^1^H-^15^N HMBC correlations for compound **1**.

**Figure 3 molecules-28-02937-f003:**
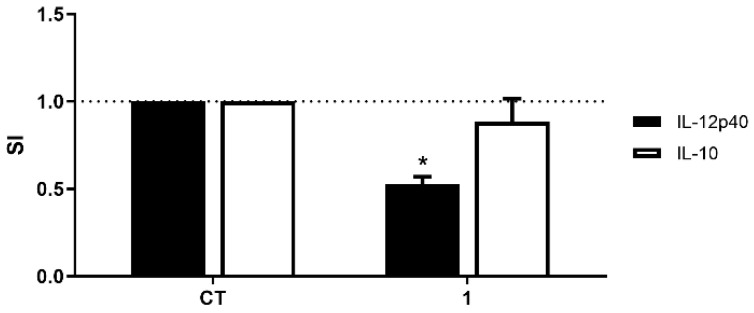
The effect of geobarrettin D (**1**) on DC secretion of IL-12p40 and IL-10. DCs were matured and activated with IL-1β, TNF-α and LPS for 24 h in the absence (solvent control (CT)) or presence of geobarrettin D (**1**) at 10 μg/mL. The concentrations of IL-12p40 and IL-10 in the supernatants were determined by ELISA. The data are presented as SI, i.e., the concentration of each cytokine in the supernatant of cells cultured in the presence of compound **1** divided by the concentration of the cytokine in the supernatant of cells cultured without compound **1**. The results are shown as mean ± SEM, *n* = 3. Different from CT: * *p* ˂ 0.05.

**Figure 4 molecules-28-02937-f004:**
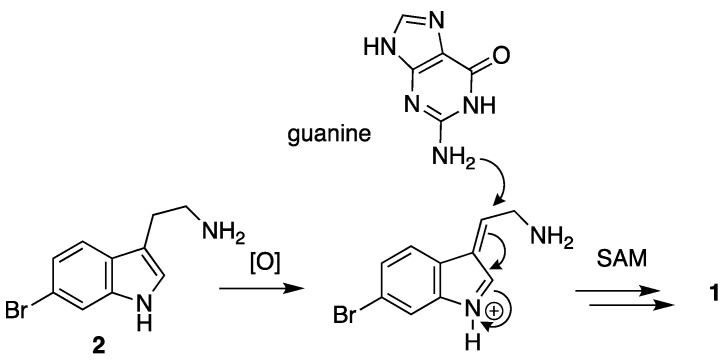
Putative biosynthesis of geobarrettin D (**1**).

**Table 1 molecules-28-02937-t001:** ^1^H NMR (600 MHz) and ^13^C NMR (150 MHz) spectroscopic data for geobarrettin D (**1**).

No.	*δ*_H_ ^a^	*δ*_H_ ^b^	*δ*_C_ ^a^	*δ*_N_ ^b^	^1^H-^13^C HMBC ^a^	^1^H-^15^N HMBC ^b^
N-1′		10.60 (1H, d, 2.0)		131.3		
2′	7.57 (1H, s)	7.53 (1H, d, 2.0)	125.9		C-1″, 3′, 5′, 3a′, 7a′	
3′			113.4			
3a′			125.2			
4′	7.65 (1H, d, 8.5)	7.12 (1H, d, 8.5)	120.8		C-3′, 6′, 3a′, 7a′	
5′	7.24 (1H, dd, 8.5, 1.6)	7.45 (1H, dd, 8.5, 1.6)	124.2		C-7′, 3a′	
6′			117.0			
7′	7.60 (1H, d, 1.5)	7.61 (1H, d, 1.5)	115.9		C-6′, 5′, 3a′	
7a′	-	-	139.1			
N-1″				47.9 ^c^		
2″	4.07 (1H, dd, 13.7, 5.9)4.15 (1H, dd, *J* = 13.7, 6.8 Hz)	4.07 (2H, m)	69.3		C-2″, 3′, -N(CH_3_)_3_	N-10
3″	6.05 (1H, t, 6.3)	6.05 (1H, t, 6.3)	45.4		C-1″, 3′, 3a′, 2′, N-10	
1″-NMe	3.30 (9H, s)	3.27 (9H, s)	54.8		C-2″, 1″	N-3″
N-1						
2			154.9			
N-3						
4			151.0			
5			110.0			
6			155.0			
N-7				156.4 ^c^		
8	9.01 (1H, s)	8.81 (1H, s)	140.1		C-11, 4, 5	N-7, 9
N-9				157.3 ^c^		
N-10				96.3 ^c^		
11	4.11 (3H, s)	3.84 (3H, s)	36.2		C-5, 8	N-7
12	3.91 (3H, s)	3.92 (3H, s)	32.1		C-4, 8	N-9

^a^ Recorded in CD_3_OD. ^b^ Recorded in D_2_O/H_2_O (1/9). ^c 15^N *δ* obtained by indirect detection from ^1^H-^15^N HMBC cross peaks.

## Data Availability

Not applicable.

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
