# Peer review of "Geobarrettin D, a Rare Herbipoline-Containing 6-Bromoindole Alkaloid from *Geodia barretti"

_molecules, 2023, doi:10.3390/molecules28072937_

Round 1

Reviewer 1 Report

I have read the manuscript submission of Xiaxia Di describing the isolation and structure elucidation of geobarrettin D, and hereby tender my findings. As could be expected from Gustaffson, Molinksi and their co-workers, this is a nicely executed piece of work. I have no significant concerns with the manuscript, bar a few minor issues and I feel it will be suitable for publication within Molecules.

1.     Line 27: I believe it should be “…shows anti-inflammatory…”; one presumes it still has this activity! (Showed could imply it no longer has anti-inflammatory activity)

2.      Line 61: IUPAC no longer recommend the terms “pseudo-molecular” or “quasi-molecular” ion; rather ions are either (de)protonated molecular ions for M+/-H, molecular ions for M+, or adduct ions in other cases. Please change this to “…exhibited molecular ion isotopomers m/z…” and see https://www.degruyter.com/document/doi/10.1351/PAC-REC-06-04-06/html?lang=en for more details.

3.       Line 64: As 1”-NMe accounts for three magnetically equivalent CH3 groups, surely only 18 signals were detected?

4.       Line 68: Highest could also imply the largest m/z value. Please change “highest” to “most intense”. Also, please show the full MS spectrum in the supplementary information file so the reader can observe this loss of NMe3.

5.       Line 84: Change lmax to lamba-max.

6.       Line 96: I found this sentence a bit difficult to interpret. I think the issue is that the molecular formula provided (C17H16BrN6O) is not the formula of geobarrettin D; please change to the correct formula (C20H25BrN7O).

7.       Lines 96 to 121: On my PDF the delta symbol for NMR chemical shifts has been converted to a little bird, which while cute, is not very helpful! I suspect that this has happened during the PDF creation process on the website but please double check.

8.       Line 104: Was a HSQC spectrum with an optimized 1JCH choice (say 200 Hz rather than the standard ~140 – 145 Hz) attempted to detect the CH-8 correlation?

9.       Line 122: Probably a similar issue, but the alpha in alphaD has become a swirl symbol. I note this has also happened in the experimental section (lines 211 – 214) so please check that lamba’s, alpha’s, nu’s have all be used correctly.

1.   I am not sure if this is standard anymore, but is it customary to include some kind of information regarding the referencing standard used for 15N NMR in the experimental section? I note no information is provided for referencing of 1H or 13C either.

1.   Were any other MNPs isolated? For example, other known geobarrettin’s? This should be noted if so.

1.   I must admit, I was somewhat surprised that what I presume is such a polar metabolite, given the salt form, was extracted in the CHCl3/CH2Cl2 partitions. Can the authors comment on this?

The authors are to be commended on the excellent quality of their spectral data provided in the supplementary information file.

Author Response

Dear Reviewer 

Kindly find attached the revised paper along with the rebuttable letter of responses

Best regards 

Reviewer 2 Report

This is a well-written article that provides an in-depth look into the indole nucleus and how it affects biological activities. The article is well organized and provides a clear overview of the chemistry and biological effects of indole and its derivatives. The article also includes a detailed description of the structure of geobarrettin D (1), a 6-bromoindole derivative, and its anti-inflammatory activities. The article is well-referenced and provides a thorough discussion of the research conducted in this area. Overall, this is an informative and well-written article that provides an insightful look into the indole nucleus and its derivatives.
Minor revision:
-This manuscript needs to have its scientific writing improved. In my brief review, I noticed writing typos and grammar mistakes, such as on lines 26-28 where the cytokine IL-10 was noted and compound 1 showed anti-inflammatory activity, and on line 159 where an "i" was added to the word "activities". It is the authors' responsibility to ensure that the scientific writing is thorough and accurate throughout the paper.
- It is imperative to provide all parameters for the HMBC and HSQC pulse programs, such as mix time, transients, pulse width, power, points, increments, sweep width, and relaxation delay.
- Moreover, it would be beneficial to include a discussion of the NOESY spectra briefly in order to compare the spatial structure of the non-steroidal anti-inflammatory drugs to that of the literature [ Journal of Molecular Liquids (2022), 367, art. no. 120502, . DOI: 10.1016/j.molliq.2022.120502].
- Furthermore, it is necessary to discuss the potential anti-inflammatory activity linked to the lipid membrane, as indicated in the literature briefly [  Molecules 2023, 28(4), 1518; https://doi.org/10.3390/molecules28041518 , Pharmaceutics, 14 (11), art. no. 2276, . (2022) DOI: 10.3390/pharmaceutics14112276].
- The  suggestion (not necessary) I have is to add more detail to the discussion of the biosynthesis of compound 1, as this is a key part of understanding the biological activities of the compound.
If the authors revise their paper based on my feedback and provide more details and explanations, I would be able to recommend it for publication in Molecules.

Author Response

(The authors gave the same response as above.)
